# DNA Origami as Emerging Technology for the Engineering of Fluorescent and Plasmonic-Based Biosensors

**DOI:** 10.3390/ma13092185

**Published:** 2020-05-09

**Authors:** Morgane Loretan, Ivana Domljanovic, Mathias Lakatos, Curzio Rüegg, Guillermo P. Acuna

**Affiliations:** 1Photonic Nanosystems, Department of Physics, Faculty of Science and Medicine, University of Fribourg, Chemin du Musée 3, PER08, 1700 Fribourg, Switzerland; mathias.lakatos@unifr.ch (M.L.); guillermo.acuna@unifr.ch (G.P.A.); 2Laboratory of Experimental and Translational Oncology, Department of Oncology, Microbiology and Immunology, Faculty of Science and Medicine, University of Fribourg, Chemin du Musée 18, PER17, 1700 Fribourg, Switzerland; curzio.ruegg@unifr.ch

**Keywords:** DNA nanotechnology, DNA origami, biosensors, optics (plasmonic and fluorescence sensing)

## Abstract

DNA nanotechnology is a powerful and promising tool for the development of nanoscale devices for numerous and diverse applications. One of the greatest potential fields of application for DNA nanotechnology is in biomedicine, in particular biosensing. Thanks to the control over their size, shape, and fabrication, DNA origami represents a unique opportunity to assemble dynamic and complex devices with precise and predictable structural characteristics. Combined with the addressability and flexibility of the chemistry for DNA functionalization, DNA origami allows the precise design of sensors capable of detecting a large range of different targets, encompassing RNA, DNA, proteins, small molecules, or changes in physico-chemical parameters, that could serve as diagnostic tools. Here, we review some recent, salient developments in DNA origami-based sensors centered on optical detection methods (readout) with a special emphasis on the sensitivity, the selectivity, and response time. We also discuss challenges that still need to be addressed before this approach can be translated into robust diagnostic devices for bio-medical applications.

## 1. Introduction

### 1.1. DNA Nanotechnology

Nanotechnology allows applications of novel materials and devices at the nanoscale level in all disciplines of science. One emerging material for nanoscale engineering is DNA [1]. Some of the main characteristics of DNA are the high specificity, programmability, flexibility, and sub-nanometer precision. These characteristics allow DNA to be used for the construction of highly precise nanostructures, as demonstrated first by Seeman, in his ground-breaking research in the early 1980s [2]. Self-assembly of DNA nanostructures started with crossovers and branched junctions [3] followed by a variety of DNA objects such as polyhedral [4], prisms [5], and buckyballs [4]. Subsequently, self-assembly of DNA was further developed into an advanced order of design using DNA crossover tiles [6]. With this strategy, sticky ends are added to DNA for creating two-dimensional (2D) [7] and three-dimensional (3D) [8,9] periodic lattices. Another more advanced single-stranded DNA tiles strategy for DNA-based construction is DNA bricks that form large DNA structures in 2D and 3D [10]. All self-assemblies mentioned above are based on a combination of “short” single-stranded DNA or pre-assembled DNA motifs. The structures assembled via these methods can be used as programmable scaffolds for the organization of different nanoparticles or molecules [11], but are still limited by the size and the complexity. In 2006, a method called DNA origami was introduced by Rothemund [12]. This method enabled an increase in complexity and the creation of larger nanostructures (in the range of hundreds of nm) [13,14]. The technique consists of a long single-stranded DNA scaffold (usually from the genome of the bacteriophage M13, containing approximately seven to eight thousand nucleotides), which is folded into desired shapes with the help of a set of hundreds of short complementary staple strands (approximately 40 nucleotides long). One of the advantages of the DNA origami method compared to other assembly strategies is the simplicity of using a scaffold strand and folding it into any pre-designed shape, and that it does not need a stoichiometric blend of component strands [15]. The scaffold has a well-known nucleotides sequence, whereas complementary nucleobases of the staples are carefully chosen depending on the wanted design of the DNA origami structure. Upon hybridization of the scaffold with the staple strands, every nucleotide of each DNA staple will then “take the right place” by hybridizing with the complementary sequence on the scaffold. Therefore, the “folding” of the scaffold into the designed specific shape takes place in solution under temperature and salt concentration conditions ensuring the complete annealing of all strands (Figure 1). This technique allows the design of highly diverse, complex, multi-functional, 2D, and 3D structures [16,17,18,19,20,21,22,23,24].

The fact that the whole sequence is known, together with the ability to attach functional molecules to single strands in DNA origami structures with a high precision and accuracy, makes DNA origami a highly addressable tool with predictable conformation structures [25]. Besides, DNA origami structures are natural biopolymers that are biocompatible, biodegradable, and minimally toxic [26]. Thus, this approach has a great potential in biomedical applications [26,27]. Furthermore, it was shown that DNA origami structures are more stable than other nucleic acid structures, facilitating their use for biological and technical applications [28]. In addition, the development of DNA origami enabled the elaboration of “dynamic” structures whose functions rely upon conformational changes induced by external stimuli, externally applied fields, or strand-displacement reactions [29]. The conformational changes may be then exploited to generate measurable signals to “sense” the original stimulus [29,30]. The dynamic structures can be used in different applications [31,32,33,34], in particular for biomolecular sensing. For example, a dynamic structure can be designed to open and close when entering in a specific environment or upon detection of a specific target. The conformational changes induced by target binding can be detected using different techniques, such as optical and electrochemical ones [22,24], that take advantage of precise positioning for signal enhancement, transduction, or implementation of logic operations. In that case, the functionalization of DNA origami structures can be performed with high-photostable fluorescence emitters such as quantum dots, metal nanoparticles (NPs), and/or fluorophores. For instance, the creation of plasmonic sensors, which enhances the fluorescence emission [35] and Raman scattering [36], is possible when plasmonic NPs are incorporated within DNA origami structures. Such structures can be composed of a heterogeneous choice of NPs like for example a silver NP sandwiched by two gold NPs [37] or a core-shell nanoparticle made with a gold core and a silver shell [38]. The creation of fluorescence sensors is achieved when one or more fluorophores (fluorescent molecules) are functionalized within a structure to create a fluorescence signal. To sum up, plasmonic and fluorescence DNA origami-based sensors are promising candidates for single-molecule studies due to their predictable conformation structures, their compatibility with biological environments, and their high detection sensitivity, with an enormous potential for translation, in particular for bio-medical applications [21].

### 1.2. DNA Origami-Based Structures Used for Biomolecular Sensing

Implementation of new sensing concepts could potentially improve standard approaches of biomolecular sensing such as polymerase chain reaction, northern blotting, next-generation sequencing, or enzyme-linked immunosorbent assay (ELISA) [39,40]. Regardless of their enormous analytical progress, these methods have certain limitations [41]. Most typically, they are complex, involve multiple steps, and are time-consuming. Therefore, there is a continuous need for improvement and development of more efficient tools. DNA origami-based technology has the potential to revolutionize the development of a new generation of highly specific, fast, high throughput analytical devices with a single-molecule sensitivity for bio-medical applications [21,42,43].

DNA origami sensors combine two main elements: (1) a recognition unit, which is responsible for the detection of the target of interest, and (2) a transducer unit, which is responsible for converting a molecular interaction between the sensor and the analyte into a measurable physical signal [41]. For example: optical, electrochemical, thermal, electrical, mechanical, surface acoustic waves or ion-sensitive mechanisms can be used to translate a molecular interaction into a physical, measurable signal [44,45]. They are able to detect different analytes such as proteins [22], small molecules [46], alterations in pH [47], ion concentrations [48], and temperature [49]. These interactions can be translated into optical signals by plasmonic or by fluorescence excitation [50,51]. Optical DNA origami sensors are developing into sensitive, rapid, accurate, flexible, and multi-target sensing tools with a low limit of detection (LOD) [18,19,21,52,53,54,55]. They are used in biology, environment monitoring, and the food safety industry, and have a great potential for applications in biomedicine to detect and monitor diseases and pathogens [56].

This review focuses on the recent progress on applications of plasmonic and fluorescent-based DNA origami sensors for biomolecular sensing with a focus on their selectivity, limit of detection, and response time. We will conclude with an outlook where challenges and potential future developments will be discussed.

## 2. DNA Origami Complexes Used in Optical Biomolecular Sensing

### 2.1. Fluorescence-Based Sensors

Fluorescence is a process that can occur in a molecule previously excited by, for example, incident light. An electron on the ground state (S_0_) can be promoted by an incident photon to a vibrational state, of S_1_, the first electronic excited state of the molecule. The electron first decays rapidly (typically in the picosecond-range) to the lowest vibrational state of S_1_. This decay is called internal conversion. After the internal conversion, the electron will relax from S_1_ to S_0_. One of the possible pathways for relaxation is called fluorescence, in which a photon is emitted. This emitted photon has a lower energy (longer wavelength) than the exciting photon because of the energy spent due to the internal conversion. The typical fluorescence lifetime is in the nanosecond-range and corresponds to the average time during which the electron stays in the excited state S_1_ before returning to S_0_. Besides the lifetime, the quantum yield is an important parameter to characterize the photophysical properties of a fluorescence-based sensing system. The quantum yield corresponds to the ratio between the number of emitted photons and the number of absorbed photons by the molecule [57,58].

Fluorophores, such as Cyanine, Alexa, Fluorescein, or Atto, are fluorescent molecules (or dyes) that can be incorporated into DNA origami-structures to form sensors, in two different ways [57]. The first method involves covalently-bound complexes in which fluorophores are covalently attached to the surfaces of DNA origami structures through staple strands or short oligomers [59] with high accuracy [59,60] and without DNA structure alteration [61]. However, this technique is expensive [59]. The second method is based on non-covalently binding of the dye to the DNA complex: through electrostatic interaction (within the external DNA double helix) [62], by binding to the minor or major groove of the DNA, and by intercalation (between base pairs) [59,63,64]. The latter allows easy and fast [65] incorporation at lower costs compared to covalently bound complexes. On the other hand, the intercalation of dyes between the based pairs induces distortions within the structure of the DNA complex [66], they cannot bind specifically [59,66,67], and in many cases this approach shows reduced efficiency and selectivity for sensing [66]. On the contrary, the interaction of fluorophores with DNA grooves causes less distortion within DNA structures compared to intercalators [66]. Fluorophores are available for different wavelengths all over the visible spectral range for both absorption and emission. This property allows the construction of sensors with a specific fluorophore, or a combination of them, to achieve a defined emission profile for the detection of single or multiple target molecules [57,68]. Generally, fluorescence DNA origami-based sensors have led to the development of three different methods for biomolecular sensing: (1) direct fluorescence emission (turn-on), (2) fluorescence (Förster) resonance energy transfer (FRET), and (3) fluorescence quenching (turn-off).

As reported by Domljanovic et al. [69], DNA origami structures, in which dyes are non-covalently incorporated, were studied to detect anti-DNA antibodies (Figure 2A). This method exploited the fact that the number of dyes (Eva Green) interacting with DNA changes when antibodies bind to the DNA-dye complex. This principle allowed the specific recognition of anti-DNA antibodies in systemic lupus erythematosus (SLE), an autoimmune disease characterized by the production of antibodies against genomic DNA. Results showed that the specificity of the DNA origami-dye complex immunofluorescence assay is comparable to the enzyme-linked immunosorbent assay (ELISA) (81–93% and 80–94% respectively) typically used in SLE diagnostics. However, the LOD (sensitivity) is 10-fold higher than for ELISA. Furthermore, the amount of serum needed is small (2 μL) and the time of assay is shorter (1.5 h) compared to ELISA (typically 100 μL and 6.5 h, respectively).

In 2019, Deams et al. [54] presented a cost-effective and sensitive enzyme-linked-hybrid aptamer immunosorbent assay to detect the peanut antigen Ara h1. Magnetic microparticles were coated with 2D DNA origami structures, each functionalized with aptamers to specifically bind at least one Ara h1 protein molecule. DNA origami structures allow controlling the aptamers surface density and optimizing the accessibility and the capture efficiency of the targets. They also decrease the number of aptamers needed to reach a low LOD. In a one-pot solution, the peanut antigen was added to the coated microparticles, followed by anti-Ara h1 antibodies labeled with the dye fluorescein di-β-D-galactopyranoside. To detect the complex, a second, enzyme-labeled antibody was added in solution, with the purpose to cleave off fluorescein di-β-D-galactopyranoside while retaining the fluorescein dye (Figure 2B). The magnetic microparticles in the solution are then loaded in a microwells array and captured with the help of a magnet. Each microwell can hold a single microparticle. Fluorescence recovery was detected by standard fluorescence microscopy. Compared to directly aptamer coated microparticles (232 fM, 45 pg/mL) a LOD of one order of magnitude lower was measured (18 fM, 3.5 pg/mL) after a reaction time of 3h 30 min. Reported LODs for ELISA and lateral flow assays were 12 ng/mL and 450 ng/mL, respectively [54]. These results showed a clear improvement in sensitivity when using DNA origami structures. Many approaches use fluorophores within immunoassays to improve the assay sensitivity, required for early disease detection (i.e., when the disease is still asymptomatic). A recent article by Rutten et al. [43], reported a bioassay to detect thrombin in only 25 min consisting of a DNA origami structure labeled with aptamers and fluorophores with a LOD of 2 ± 0.2 nM. Chen et al. [70] developed a sensor using an alternative DNA origami method called DNA nanoribbons. Instead of the addition of hundreds of staple strands, as used to form standard DNA origami structures, they incorporated a maximum of three staple strands per single scaffold formed by rolling circle amplification. Fluorophores carboxyfluorescein (FAM) were then incorporated within DNA nanoribbons to sense intracellular pH. The results showed that the sensor had a desired sensitivity response for pH in a range of 4.0 to 8.0 in Hank’s balanced salt solution. In endosomes, a pH-decrease from 6.4 to 5.6, was measured for 30 min. The authors reported that each measured change of the endosome pH corresponded to its maturation steps. As the endosome function is to transport materials from the outside to the inside of the cell, the DNA nanoribbons could also be used for delivering siRNA to the cytoplasm for gene silencing purposes, for example in cancer cells (load capacity of 45.7 wt %).

Numerous achievements in fluorescence-based DNA origami sensors are summarized in Table 1. Overall, they can reach a limit of detection in the sub-nanomolar range. The enhancement of fluorescent signal may enable the use of more affordable and portable detectors, e.g., smartphones [71,72]. One of the strategies for signal enhancement is to include many fluorophores on a single DNA origami structure. The company Gattaquant has developed fluorescence intensity standards, so-called Gattabeads, based on this approach. A second, alternative strategy is based on the local enhancement of the electric field near the surface of noble metal nanoparticles [73]. A more detailed explanation of this second strategy is given in the surface-enhanced Raman scattering section.

### 2.2. Fluorescence (Förster) Resonance Energy Transfer (FRET)-Based Sensors

DNA origami-based sensors can exploit the fluorescence (Förster) resonance energy transfer (FRET) as a method for sensing. This method can also be used to study molecular interactions, conformational changes (open-close structures), single-molecule detection [57,68,74], or as spectroscopic rulers [75]. FRET occurs between two fluorophores whereby one acts as a donor and the other as an acceptor. It is a non-radiative process generated by the dipole-dipole interaction between the two dyes [57]. In other words, when two dyes are in close proximity (typically under 10 nm), the excited donor can transfer energy to an acceptor in the ground state and excite it. Once the acceptor is excited, its electron can relax from the excited state to the ground state resulting in fluorescence emission [76,77,78]. To achieve FRET a spectral overlap between the donor emission and the acceptor absorption and an optimal distance between fluorophores are key parameters to consider. The FRET efficiency is proportional to the inverse sixth power of the donor-acceptor distance and it typically covers the 2–10 nm range [78]. The efficiency is also dependent on the relative orientations of the donor and acceptor pair. However, it is worth noting that fluorophores cannot be deterministically oriented in DNA-origami structures but are either free to rotate or fixed in an uncontrolled orientation depending, among other factors, on the fluorophores charge.

Since the fluorophores can be precisely arranged on a DNA origami structure, assembly of multiple dyes exhibiting FRET is possible. An accurate positioning of many fluorophores has been used to study multi-dyes FRET cascades and homo-FRET [76,79]. Linear FRET arrays have been further developed to form multi-targets sensing complexes [78]. With the development of more complex and dynamic DNA origami-based structures, new sensing strategies are implemented by including light-harvesting and signal amplification [80]. One of the first designed, constructed, and characterized FRET-based DNA origami sensors was reported by Andersen et al. [18] in 2009. They designed a 3D DNA origami box consisting of six origami sheets and a controllable lid operated with a dual lock-key system composed of DNA duplexes (Figure 3A). The DNA duplexes provide a ‘toehold’ for the displacement by externally added oligonucleotides called ‘keys’. The toehold mediated strand displacement (TMSD) is based on the dislocation of one strand of DNA or RNA, previously bound to a partially complementary substrate strand, by a soluble, fully complementary DNA or RNA oligonucleotide, a fundamental principle in nucleic acid nanotechnology [81]. To detect the opening of the box lid, two fluorescent dyes, Cy3 and Cy5 (acting as donor and acceptor, respectively), were incorporated in the opposite surfaces of the box. The addition of ‘keys’ opened the box, therefore changing the FRET (due to a change in the distance between fluorophores) and making the interaction measurable (Figure 3C). Overall, they validated a DNA origami structure that can sense a 200 μM concentration of targets within ∼40 s upon their addition. Concomitantly, the box has the potential to store cargo and release it upon opening. Another example of a DNA origami box with the possibility of executing at least three cycles of complete opening and closing was reported by Zadegan et al. [24]. The first opening of a box with 0.4 μM of keys was efficient in the range of minutes, while the reopening took 4 h. This work demonstrated the feasibility of sensing two-breast cancers miRNAs, miRNA223 [82], and miRNA30c [83]. In addition, work done by Grossi et al. [84] improved the storage, the selectivity, the specificity, and the response time of a DNA origami box called DNA nanovault (DV). The opening and closing mechanism of the DNA origami structure was designed to be recurrent multiple times and the response time was reported in the range of minutes. One more example worth mentioning is Ijäs et al. [85], they reported a DNA origami nanocapsule with a high pH sensitivity, an extremely fast opening (30 s) as well as a possible design of smart DNA origami-based system that responds to a stimulus in living organisms.

To detect proteins instead of miRNA, the DNA origami box was modified to include aptamers [86,87]. The results showed that the DNA origami devices could detect up to 100 nM of protein with a response time of about 20 min. Strand displacement by the target [88] coupled to the force present in base-stacked DNA duplexes [89] is the most common way to activate the opening of a DNA origami complex. However, these techniques are limited to the nM concentration range and the response times are in the range of minutes or longer. Subsequently, DNA origami structures with response times of seconds or less were reported. Marras et al. [90] used the hybridization of sequence-specificity DNA for changing the configuration of the structure in response to environmental stimuli (Figure 3B). The structure was designed with short ssDNA sticky ends, which can rapidly hybridize or dehybridize in response to changes in concentration of cations in solution, resulting in a conformational change of the structure. A single pair of FRET fluorophores was incorporated within the complex to detect the opening and closing of the structures [90]. Conformational changes within a DNA structure could be triggered with different types of ions including mono-, di-, and trivalent cations. Single-molecule FRET measurements showed that closing and opening can take place within the millisecond range (≤ 200 ms) and can be repeated within one second. Thus, rapid control of DNA origami conformation with different ions [48,91], broadened the range of activating mechanisms, and revealed the power of DNA origami complexes in sensing environmental cues.

The change in pH of solutions was used as a trigger for sensing purposes. In that case, the incorporation of a pH-sensitive dye pair that exhibit FRET within the DNA origami structure (ratiometric pH nanosensors) was used. The sensing principle behind these nanosensors is the modification of fluorescence intensity by the change of pH of the solution [80]. A rise in pH from 6 to 8 resulted in the decrease of the donor fluorescence lifetime detectable in only 5.71 ns. This work showed that besides the rapid activation, the arrangement of dyes at specific positions within a given structure and their number are important parameters affecting the sensitivity of the complex [55,80]. An external applied stimuli could then rapidly trigger DNA origami responses. For instance, an applied electrical potential (voltage above 200 mV) can be converted into optical signals at the single-molecule level within a few seconds only [92]. A further example of the development of fast, selective, and sensitive sensors was reported by Selnihhin et al. [21]. They designed a “book”-like structure where donor fluorophores were placed on one side and acceptor fluorophores on the other side of the structure. Thus, in the closed state, fluorophores were in sufficient proximity to allow FRET. The addition of a fully complementary DNA keys resulted in the opening of the sensor and a decrease of the FRET signal. The structure could detect target sequences down to 10 pM concentrations with a detection time of 2 min. Beside high sensitivity, selectivity, and rapid time of detection, this work demonstrates the possibility to construct multiplex sensors to detect several targets based on different dye emission profiles. In short, the emerging different possibilities of fast triggering mechanisms of FRET DNA origami-based structures (Table 2), and the possibility of sensing at the single-molecule level [92,93], are unlocking previously unsuspected possibilities for using DNA origami as a sensor nanodevice.

### 2.3. Quenching-Based Sensors

While some sensors use specific emission profiles of fluorophores to detect a molecule of interest, others use the decrease of their overall fluorescence intensity (turn-off), a process known as quenching. Many different processes can induce quenching. Quenching can be static, dynamic, or a mixture of both. The static quenching is due to the formation of a ground-state complex between the fluorophore and quencher pair. In that case, when the complex absorbs a photon, it returns immediately to the ground state without fluorescence emission [94,95]. The main characteristics of static quenching are changes in the absorption spectra of the fluorophores and an absence of changes in their fluorescence lifetime (as the observed lifetime comes from fluorophores that do not form a complex with a quencher (i.e., free fluorophores) [57]. In the case of dynamic quenching, also called collisional quenching, the quenching process occurs only when the fluorophores are in the excited state (i.e., the fluorophores have absorbed the incident photons) [57,94,95]. In that case, the lifetime is reduced as an additional quenching rate induces a depopulation of the excited state [95,96]. Dynamic quenching allows faster and more reversible detection than static quenching [90]. Despite this, DNA origami-based sensors could use dynamic, static quenchers, or quenchers with both properties. In order to know which type of quenching is dominant, the spectrum and the lifetime of the dye-quencher pairs must be measured [57,95]. Overall, combining DNA origami with quenching as a method of detection opens the possibility of multiplexing. Precise positioning of fluorophores and quenchers within a structure gives the ability to construct barcodes of targets by choosing unique fluorophores to obtain specific emission profiles, which could permit the detection of several different targets.

Ke et al. [97], reported a DNA origami nanoactuator consisting of four DNA origami joists linked into a diamond shape via flexible ssDNA joints. The mechanism of changing conformation can be controlled in two different ways by the addition of extra strands. The first way is called ‘strut-lockings’ and is formed by two rigid DNA double helices, the connecting strut. The second one is called ‘corner-locking’ and uses DNA strands only. In the presence of the corner-locking strands, the DNA origami nanoactuator switched from closed to open state, resulting in the change in the fluorescence signal of the 6-FAM/BHQ-1 fluorophore-quencher pair. This mechanism responded to a broader range of triggers for its opening. The first demonstrated trigger for the opening mechanism was a change in the buffer solution in the presence of 100 mM KCl. The second trigger tested was endonuclease enzymatic activity (i.e., 5 units of the restriction enzyme BamHI) in 100 mL of 20 nM closed conformation DNA structure during a 10 min incubation. The third trigger tested were oligonucleotides: 100 nM DNA nanoactuator were incubated for 2 h with miR-210 at 15 equivalents. In addition, authors repurposed and redesigned the structure by the addition of extra strands, called ‘strut-locking’, thereby creating an allosteric activation mechanism. The nanoactuator structure could change its conformation very accurately just by adjusting the length of the connecting struts. The change of distance induced in one part of the structure would be mirrored by the other part, through large-scale movements. The specificity and sensitivity of this device was pushed to the single-molecule level to monitor weak interactions. This work showed the possibility of multi-purposing single devices as a platform for mobility shift study [98] and as a platform for the detection of different targets at the single-molecule level. Further improvements could be to increase the sensitivity and the detection time of the complex.

Sensitive diagnostics of diseases such as cancer are essential for early detection and efficient patient treatment and monitoring [99]. Standard approaches of biomolecular sensing still need an improvement, especially of their robustness and reproducibility. Recently, a branched DNA technique was introduced for molecular diagnostics [100]. This method enhances the signal through a cascade of hybridization reactions. In 2017, Ochmann et al. [101] reported a possible diagnostic approach based on a direct physical fluorescence amplification method. This approach involved the creation of DNA origami complexes with fluorescence quenching hairpins, and plasmonic nanoparticles that acted as a so-called optical nanoantenna [102,103]. In this study, the DNA origami complex consisted of dye-quencher pairs (ATTO 647N dye and a BlackBerry quencher 650) in close proximity when the hairpin was closed, resulting in reduced fluorescence signal (Figure 4A). In the presence of a complementary oligonucleotide target, the hairpin opened up and the fluorescence intensity increased. They used this platform to detect a synthetic Zika-specific target (artificial DNA and RNA) at 1 nM concentration range in a standard buffer and human serum after 18 h incubation. This concept showed a stronger signal enhancement at the single-molecule level that could lead to improved, highly sensitive, portable biomolecular assays.

Another approach of signal enhancement is based on a linear amplifier such as the DNA walker (Figure 4B). A DNA walker mimics natural molecular motors by working on the principle of Brownian motion and chemical energy [104]. DNA walkers have been significantly perfected in recent times [105] due to the improvement of biophysical tools [104,106]. The incorporation of molecular motors in DNA nanostructure is now possible thanks to the driving force given by the extra base pairing from fuel oligonucleotides. These molecular motors can be used as building blocks in nanotechnology [91] or for amplified detection of oligonucleotides [104]. Wang et al. [107] created a complex of DNA walkers with a nicking enzyme [105,108] to enhance the fluorescence signal. In this work, the DNA walker triggers an amplification reaction by walking on a rectangular DNA origami structure, thereby liberating quenching molecules and releasing space for fluorescence labeled imager strands (Figure 4B). The DNA origami walker complex is designed with single nucleotide sensitivity and can detect up to three mismatches within the sequence. As well, it can make from 20 to 60 steps. Since walking was based on toehold-mediated migration, the length, and the binding efficiency of the target, its speed of walking depends on the number of mismatches. Nevertheless, results showed a decrease in the fluorescence intensity with respect to an increase of mismatches within the targets. The generated signal is significantly higher relative to the background signal, which makes these devices one-step closer to possible applications in biomedical biosensing. Another technique that exploits quenching is presented in the articles of Kroener et al. [109,110]. They fabricate rod-shaped DNA origami structures, which can be oriented by an applied external electric field. The main working principle is as follows: DNA origami structures are bound through one end onto an electrode. Since the DNA origami structures are negatively charged, the angle at which these structures stand can be adjusted by varying the charge (sign and amplitude) of the electrode. The overall distance between the free end of the rod-shaped DNA origami and the electrode is monitored by labeling the free end with a fluorophore and studying the distance dependent fluorescence quenching induced by the electrode. An overview of some quenching method for sensing together with DNA origami structures is summarized in Table 3.

### 2.4. Surface-Enhanced Raman Scattering-Based Sensors

Compared to fluorescent molecules, nanoparticles (NPs) possess higher stability under physiological conditions as they do not show photobleaching, blinking, and lifetime limitations [111]. That is why their optical properties are exploited for sensing. Moreover, the interaction of NPs with light is strongly dependent on the NPs size and material together with the light frequency, among other parameters. If the NPs have a size much smaller than the wavelength of the incident light, the electric field penetrates the NPs leading to a displacement of the NPs´ free electrons, causing a polarization of the NPs. This polarization in turn leads to a restoring force. If the incident light has an oscillating frequency that matches the eigenfrequency of the NPs, a strong collective electron oscillation (known as localized surface plasmons) arises [112,113]. The oscillation frequency strongly depends on the size, the shape, the type of material of the metallic NPs, and the dielectric environment [112,113]. For gold and silver NPs, these resonances are mostly located in the visible range of the optical spectrum. Sensing applications based on gold and silver resonances can benefit from the ability of the DNA origami method to precisely arrange and align nano-objects in a highly ordered manner at the nanoscale level.

One of the methods allowing ultrasensitive detection with the potential to characterize or detect structures up to the single-molecule level, is surface-enhanced Raman spectroscopy (SERS). SERS is based on the scattering process of photons interacting with molecules and/or atoms in closer proximity to the metal structures, e.g., rough metal surfaces or metal nanoparticles. The standard Raman signal is increased by several orders of magnitude (e.g., 10^6^ to 10^7^ for single spherical Ag, Au NP) due to the local field enhancement effect after plasmon resonance excitation of the metal structure (Figure 5A,B) [114]. A local concentration of the field, e.g., by changing the curvature of the metal structure (from rounded-shape to needle-like) or special NPs arrangements will further increase this effect (Figure 5A,B) [115].

The DNA origami technique allows precise positioning of metal nanoparticles. In particular, for SERS applications, the aim is to reach the shortest gap between the nanoparticles. One of the first examples based on this approach was reported by Prinz et al. [36]. They used two spherical gold nanoparticles (AuNPs) assembled on the surface of a triangular-shaped DNA origami structure and the Raman signal generated by a specific number of TAMRA fluorophores, to optimize particle size and gap [36]. Similar fundamental studies on the SERS effect involving DNA origami nanostructures were described by Thacker et al. [116] and Kühler et al. [117]. Simoncelli et al. [53] reported a complex DNA origami platform for SERS with dyes located precisely at the hot spot region of the NP dimer. By thermal shrinking of the DNA origami structures (1–2 nm), different gap-dependent SERS signals (close to the single-molecule level with an adjusted number of one or four Cy3 and C3.5 fluorophores) could be measured [53]. Beside dimer structures with spherical nanoparticles, other nanoparticle geometries like stars [118] or triangles [119] in combination with DNA origami structures were investigated. Tanwar et al. [118] studied the SERS signal of a single Texas red dye placed within the gap of two Au nanostars, whereas Zhan et al. [119] showed results for SERS measurements at the single-molecule level for Cy5 and Cy3 molecules. Compared to standard measurements in solution, these structures enhanced the SERS signal by a factor of 10^9^ to 10^10^. Further improvement in the sensitivity of single-molecule SERS detection was achieved by the incorporation of an extra modification layer on top of the plasmonic assembly [120], or a silver (Ag) layer grown on the AuNPs dimer placed on a triangular DNA origami structure [121]. Intense SERS hotspots were created with a Raman enhancement up to 10^10^, which allowed the detection of single molecules of Cy3 and TAMRA placed at the dimer center [121]. Also, the incorporation of silicon nanowires SiNWs demonstrated a sensing improvement at subwavelength spatial precision [122]. There are several strategies under development at the moment showing a high potential to further improve the SERS sensitivity, like using a higher number of NPs (three to four), different arrangements of NPs [53,120,123,124,125] or data analysis in combination with special algorithms such as the inverse quantum (Spectral-IQ) [126]. To summarize, nanostructures of DNA origami with metal nanoparticles provide new opportunities for fast sensitive detection at the single-molecule level, as well as control of photobleaching, addressability of different sizes of nanoparticles, and signal enhancement. Overview of development of SERS DNA origami-based sensors can be found in Table 4.

### 2.5. Circular Dichroism-Based Sensors

Another spectroscopy method in combination with DNA nanotechnology that gained attention in recent years is the so-called circular dichroism spectroscopy (CD). CD gives detailed information about the chirality of a molecular structure; the chirality is derived from the lack of a center of symmetry [127] or mirror-symmetry planes of the structure [127,128]. This results in different absorption behavior for left- or right-handed polarized photons and therefore different CD signals [19,127,129,130]. For chiral molecules or biomolecules, this signal is typically strongest in the UV-region of the optical spectra, e.g., by a sensitivity for proteins in the order of 0.1 mg/mL [131]. Here, a significant enhancement of the sensitivity of CD spectroscopy by several orders of magnitude can be realized in combination with plasmonic DNA nanostructures [127,131]. Kneer et al. [132] reported on the detection of the B-form conformation of a DNA origami structure within the hotspot region of two spherical NPs. These nanoantenna structures (at 100 pM) enhanced the CD signal 30 times compared to measurements of the pure DNA origami structures (1 nM). Additionally, the signal is shifted from the UV to the visible range of the optical spectra corresponding to the plasmon resonances of the nanostructures.

Another approach in combination with plasmon-enhanced CD spectroscopy is to use DNA origami as templates for the assembly of intrinsic plasmonic-chiral structures. They can be designed to act as static or dynamic structures. Static DNA origami structures usually consist of a DNA origami structure carrying small AuNPs organized into a left-handed (LH) or right-handed (RH) helix [129,133]. So far, this or similar approaches were mainly used to examine chiral structures [129,133,134]. However, in terms of sensing and qualitatively detection of targets, dynamic acting plasmonic DNA nanostructures are more promising. Different dynamic designs were proposed, including tetrahedron metamolecule [135], a DNA-guided plasmonic helix [136], or the two-arms metamolecule [19,20,52,137,138]. Nevertheless, to the best of our knowledge, only the two-armed metamolecule was used for sensing (Figure 6). This typical DNA origami-based CD sensor was invented by Kuzyk et al. [139]. It consists of two arms each carrying a gold nanorod, connected via a flexible pivot point. By rotation, a left-handed or a right-handed chiral structure can be formed. DNA oligonucleotides on the edges of the DNA origami arms, serve as locks through the recognition of a specific target [139]. These two-armed structures can detect two different targets, each of them locks the structure in a respective chirality. Moreover, two reopening keys, one for each target, can reset the sensor. The LOD was in the range of ~70 nM.

Thereafter, this approach was extended to several different physico-chemical variables or molecules, like pH [20], light [140], thrombin [52], adenosine [137], or viral RNA [19]. Especially, for the RNA of hepatitis C virus genome a LOD of 100 pM in a buffer and 1 nM target RNA in 10% human serum was obtained with a response time of 30 min for an optimal signal [19]. Similar structures published by others have been used to build up devices for the detection of organic molecules. Zhou et al. [138] created a reversible dynamic DNA origami-based structure with the ability to sense cocaine and adenosine triphosphate in one single device. The reversibility of the adenosine-cocaine sensor is induced by a rise of the temperature. Obtained results demonstrated that sensing of 1 mM of adenosine and cocaine could be done with a LOD of 50 μM (2 adenosine molecules/lock) and 20 μM (1 cocaine molecule/lock), respectively. Furthermore, Zhou et al. showed that the length of the hybridization segment for adenosine was important to increase the CD signal, opening the possibility for further improvement of circular dichroism-based sensors. A short overview of CD-based DNA origami sensors can be found in Table 5.

## 3. Conclusions and Perspectives

In this review, we outlined the spectacular development in the field of optical sensing based on DNA origami nanostructures. The progress made allowed better complex designs for the detection of molecules of interest in the concentration range of nanomolar to picomolar [21]. Nevertheless, some more challenges lie ahead. Improving the limit of detection of the devices in the sub-picomolar range would provide new opportunities for detecting new targets as new biomarkers and for pushing the sensitivity of the already known targets to lower concentrations, possibly at the single-molecule level. This may be particularly relevant for the detection of cell-free, tumor-derived DNA (ctDNA), mRNA (ctmRNA), micro RNA (miRNA), viral DNA or RNAs, proteins, or small molecules [141]. New approaches to improve sensing are already being tested, such as plasmonic optical antennas based on NP dimer structures or the use of multiple fluorophores for signal enhancement. Rapidness of the sensors is another key issue when considering biosensors for biomedical diagnostic purposes. Usually, low target concentration induces slower response times. Nowadays quantitative detection in the micromolar or nanomolar ranges requires analytical time in the hours. Improved (shorter) response times of sensors will be key for the development of DNA origami sensors for biomedical applications. Moreover, combining DNA origami sensors with different proteins such as enzymes would possibly improve the DNA origami complex functionality [22,142]. The improvement of super-resolution imaging could possibly help with quantitative target detection in vitro and allow studying interactions of a few different biomolecules in situ [38]. A further important point is the stability and performance of dynamic mechanisms under physiological conditions, such as in biological fluids (blood, plasma, serum urine, saliva, cerebrospinal fluid). More and more studies are focusing on how to improve stability by redesigning the structure with different types of coating (polyelectrolyte [143] or encapsulation of DNA origami structures [144]). As many of these stabilization strategies influence the target binding and as well on the dynamic performance of the structures, the quest for improved stability may prove difficult. Furthermore, specific target detection may be perturbed by non-specific interactions, resulting in a false signal [145,146]. DNA origami-based sensors may provide new avenues to address some of these challenges as they can sense at the single-molecule level and they can become relatively stable in biological fluids, provided a stabilization method is incorporated within the complex [144]. Moreover, DNA origami-based sensors can distinguish between specific and non-specific interactions [147]. In addition, large-scale production, and storage of DNA origami sensors at affordable costs are important factors when considering biomedical applications. Indeed, DNA origami can be produced at larger scales, making this a huge advantage of this method. Several studies have already discussed the expected cost of future large-scale production of these structures and their future use in clinical studies [148,149]. Estimation of manufacturing cost would reach ten euros per dose for DNA origami-based complexes, to this it should be added additional cost: such as purity, batch consistency, and other pharmaceutical costs [149]. Just to compare to market-based prices of antisense oligonucleotide treatments, which cost up to USD 125,000 per injection [149]. However, expectations of market-based prices for the DNA origami method are lower than for antisense treatments due to large-scale production and to the possibility for treatments of several diseases. DNA origami can be combined with different methods such as lithography (DALI) [150]. This particular method combines the versatility of DNA origami with a lithography technique to create metallic nanostructure with plasmonic properties [150], which enables the manufacture of big plasmonic metasurfaces with small (~10 nm) sizes. Lithography can also be used to place DNA origami structures at a specific position as shown by Gopinath et al. [151] where a triangular shape DNA origami structure was positioned inside a photonic crystal. This technique could be interesting to further explore label-free single-molecule detection. Importantly, several studies already reported the possibility for long-term storage (i.e., up to years) without affecting the structural and functional integrity of DNA origami complexes [28,152].

In conclusion, the advancement of the DNA origami method for developing new ultrasensitive detection tools is moving very fast. These types of sensors are pushing the limits of human creativity and science since the need for this type of device is remarkably high. The possibility of rapid detection of specific targets at the single-molecule level may open unanticipated opportunities for DNA origami-based technologies to explore uncharted diagnostics fields with unprecedented precision.

## Figures and Tables

**Figure 1 materials-13-02185-f001:**
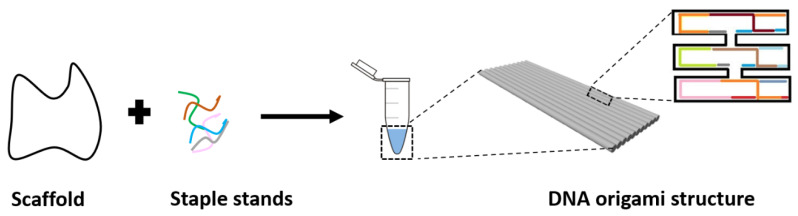
Principle of the DNA origami assembly. The DNA origami method consists of a long single-strand DNA (“scaffold”) and several hundreds of short ssDNA strands (“staples”). During an annealing process, the “staples” fold the “scaffold” into 2D or 3D structures.

**Figure 2 materials-13-02185-f002:**
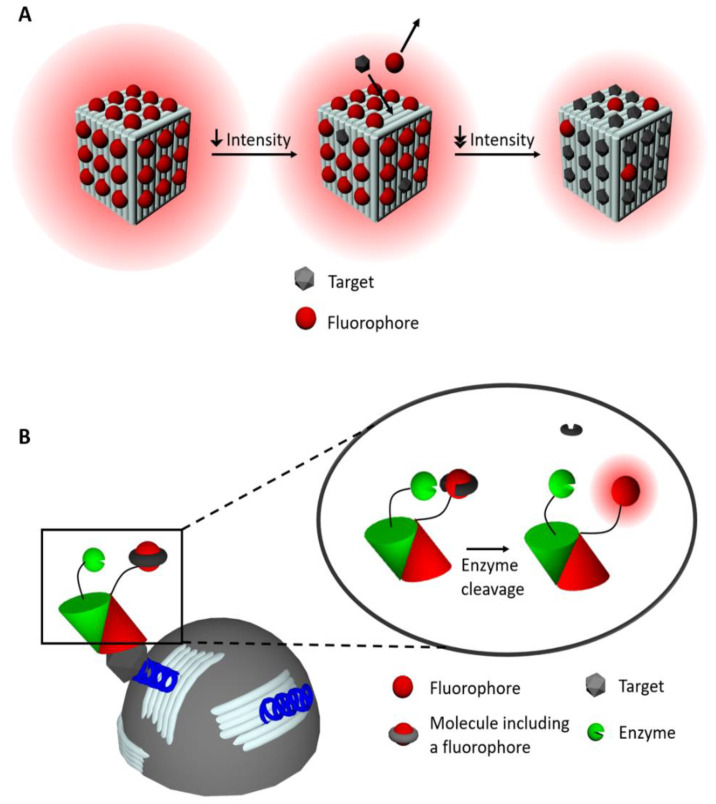
Illustrations of fluorescence sensing with DNA origami structures. (**A**) Schematic representation of the DNA origami structures (light grey) and the working principle of Domljanovic et al. [69]. The structure is bright at the starting point. In the process of detection, the fluorophores are replaced by the target, which results in a decrease in the intensity of the structure. (**B**) Schematic representation of a metallic microparticle (grey sphere) coated with DNA origami structures (light blue). The inset shows the signaling principle when the target is detected [54]. The blue helices are aptamers to which the target can bind. The red cone represents the antibody, which carries the fluorescein di-β-D-galactopyranoside and the green cone represents the antibodies carrying the enzyme.

**Figure 3 materials-13-02185-f003:**
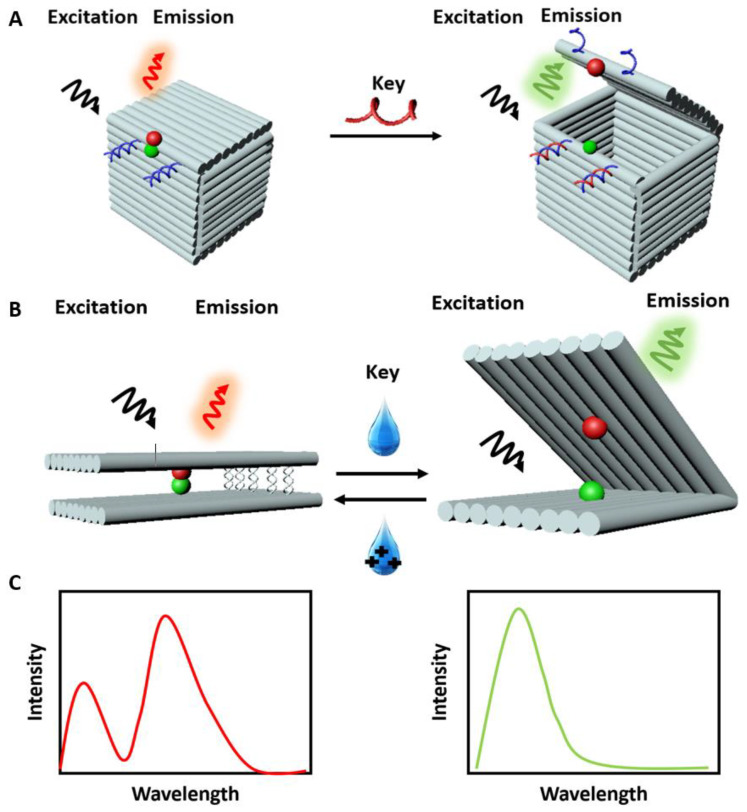
Illustrations for fluorescence (Förster) resonance energy transfer (FRET) sensing with DNA origami structures. (**A**) Representation of the sensing principle of the DNA origami box [18]. At the starting point the structure is closed (FRET occurs). The addition of a key (target) unlocks the structure and induces a conformational change. (**B**) Representation of the FRET signal of a closed DNA origami structure (FRET occurs) [90]. In the presence of cations in solution, the structure opens up (FRET cannot occur). (**C**) Representation of FRET emission spectra of closed structure (FRET occurs: red spectrum) and open structure (FRET cannot occur: green spectrum).

**Figure 4 materials-13-02185-f004:**
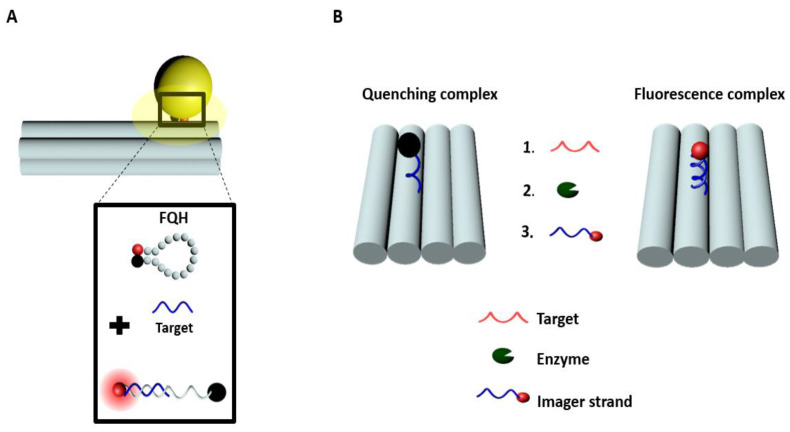
Illustrations of sensing by quenching with DNA origami. (**A**) Representation of a DNA origami pillar [101] where a fluorescence-quenching hairpin (FQH) is incorporated. Upon the addition of the target, the FQH was opened, resulting in fluorescence. To enhance the fluorescence coming from the FQH a nanoparticle was incorporated within the DNA origami. (**B**) Representation of a DNA walker [107]. At the starting point, the DNA complex is not emitting fluorescence due to the presence of the quencher. The process of fluorescence amplification/emission starts with target binding (Step 1), followed by the addition of an enzyme (Step 2) that cleaves the duplex (stator strand + target) at a specific position and finally the binding of an imager strand that fits into this position (Step 3).

**Figure 5 materials-13-02185-f005:**
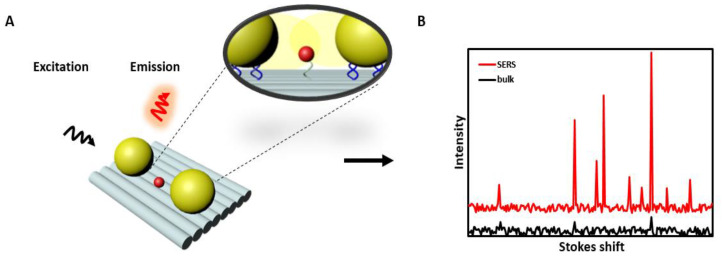
(**A**) Illustration of two nanoparticles (NPs) that are bound to a DNA origami template. When illuminated at the proper wavelength, a plasmonic “hot spot” is formed at the gap between the NPs that can be used for enhancing Raman signals. The inset shows the covalent attachment of NPs and fluorophore to the DNA origami complex. DNA functionalized NPs normally completely covered with single strands all around the surface (for clarity of the figure, these binding strands are not represented). (**B**) Representation of an exemplary surface-enhanced Raman spectroscopy (SERS) spectra (arbitrary numbers) where the Raman signal is enhanced.

**Figure 6 materials-13-02185-f006:**
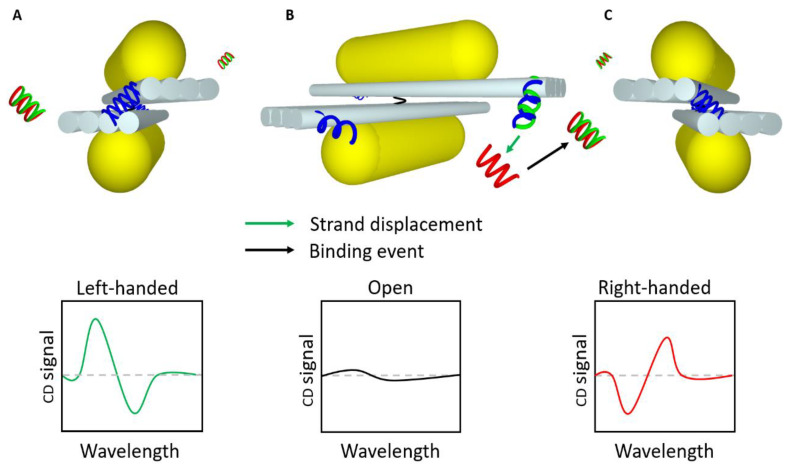
Illustration of DNA origami-based circular dichroism (CD) detection mechanism invented by Kuzyk et al. [139]. (**A**) Closed structure in left-handed configuration. (**B**) Opened structure. (**C**) Closed structure in right-handed configuration. The blue helices are the locks, the green helices are the blocking strands, the red helices are targets and the black helix between the two DNA origami sheets represents the pivot point. The graphs are sketches of CD signals typically obtained.

**Table 1 materials-13-02185-t001:** Summary table of articles using DNA origami structures (DNA-OS) with direct readout of fluorescence emission. The star (*) in front of “Signaling procedure” means a single-molecule fluorescence readout (ability to sense the fluorescence from a single DNA origami structure). Another abbreviation: Background (BG).

Signaling Procedure	Analyte	Sensitivity	Response Time	Publication Year	Reference
*Fluorescence enhancement	Single dye with 200 mM of NiCl_2_	25 μM BG fluorophores	Second range	2015	[73]
Change in pH (pH-sensitive dyes)	Intercellular pH	High sensitivity for pH from 4 to 8	30 min (pH change 6.4–5.6)	2015	[70]
Exchange of dye with target	Antibodies (systemic lupus erythematosus)	10× higher (than ELISA)	1.5 h	2017	[69]
Exchange of dye with target	Antibodies (systemic lupus erythematosus)	7% false positives (lower than ELISA)	1.5 h	2018	[62]
Enzyme cleaving	Peanut antigen Ara h1	232 fM (Aptamer) 18 fM (Aptamer + DNA-OS)	3 h 30 min	2019	[54]
Aptamer binding + labeling dyes	Thrombin	2 ± 0.2 nM (Aptamer + DNA-OS) 22 ± 3 nM (aptamer)	25 min (fluorescence record)	2020	[43]
*Fluorescence enhancement	ssDNA (Oxa-48)	2 nM	2 h (incubation)	2020	[72]

**Table 2 materials-13-02185-t002:** Summary table of approaches using DNA origami structures with FRET readout. The star (*) in front of “Signaling procedure” means a single-molecule fluorescence readout (ability to sense the fluorescence from a single DNA origami structure). Abbreviations: Open and close states (O/C), proof of principle (POP), Plasmodium falciparum lactate dehydrogenase (PfLDH), DNA nanovault (DV), time that takes to reach 50% of FRET (t1/2), Adenosine triphosphate (ATP).

Signaling Procedure	Analyte	Sensitivity	Response Time	Publication Year	Reference
O/C (strand displacement)	“Key”ssDNA (POP)	200 μM	40 s	2009	[18]
*Enzyme-assisted movement	ssDNA + Zn^2+^	1:1 (molar ratio complex: ssDNA) 1 mM (Zn^2+^)	3 nm/min (50 cleavage steps)	2010	[91]
O/C (strand displacement)	ssDNA	0.4 μM (10× molar excess)	Min range (1st opening) 4 h (2nd reopening)	2012	[24]
*Conformation Change	MgCl_2_, Temperature (POP)	5–25 mM (MgCl_2_ concentration range) 11 to 47 °C (during O/C)	-	2015	[48]
Change of telomeric DNA into guanine quadruplexes	Na^+^, K^+^	1 mM K^+^, 25 mM NaCl	-	2016	[55]
Conformation change (strand displacement + adhesive force)	ssDNA (POP)	-	-	2016	[89]
O/C (strand displacement)	ssDNA (POP)	1:1.3 molar excess (DV + closing lock) 1:1.5 molar excess (DV + opening lock)	15 min	2017	[84]
O/C (split aptamer)	ATP	0.10–1.00 mM (Range of sensitivity)	15–25 min (Observation of the fluorescence)	2017	[86]
*Interaction with environment	Depletion force	~100 fN (Resolution)	ms range (Unspecified)	2017	[93]
*Electric potential change	Optical voltage change	200 mV (Minimum before to be sensitive)	∼50 s	2018	[92]
*O/C (strand displacement)	ssDNA	10–100 pM	100 s (Efficiency with t1/2)	2018	[21]
*O/C by environment change	Cation	∼200–1000 mM (Monovalent ions) ∼5–40 mM (Divalent ions) ∼0.06–0.14 mM (Trivalent ions)	≤ 200 ms (O/C Transitions)	2018	[90]
Conformation change (strand displacement)	ssDNA	POP	POP	2018	[88]
O/C (aptamer)	PfLDH (protein)	100 nM	0–20 min	2018	[87]
Change in pH (pH sensitive dyes)	pH	6–8 (pH range to be sensitive)	-	2018	[80]
O/C (pH-latches)	pH	0.5 pH	30 s (opening) Hours (closing)	2019	[85]

**Table 3 materials-13-02185-t003:** Summary table of methods using DNA origami structure with indirect readout through quenching (turn-off). The star (*) in front of “Signaling procedure” means a single-molecule fluorescence readout (ability to sense the fluorescence from a single DNA origami structure). Abbreviations: Open and close states change (O/C), proof of principle (POP).

Signaling Procedure	Analyte	Sensitivity	Response Time	Publication Year	Reference
Hybridization of target	ssDNA	20 pmol	1 h (incubation)	2014	[99]
O/C (change in environment)	K^+^, miR-210 (miRNA) BamHI (Enzyme)	100 mM KCl 1 equivalent miR-210 (equivalents to the locking strand) 5 units of BamHI	2 h (miRNA) 10 min (BamHI)	2016	[97]
*O/C (hairpin + optical antenna)	ssDNA	POP	POP	2017	[34]
*O/C (hairpin + optical antenna)	Zika DNA/RNA	1 nM (RNA and DNA)	18 h	2017	[101]
*Enzyme-assisted moment of complex (molecular motors)	ssDNA (with mismatches)	0, 1, 2, or 3 mismatches	2 h (no Mismatch) + 4 h (with Mismatch)	2017	[107]

**Table 4 materials-13-02185-t004:** Summary table of articles based on DNA origami structures for surface-enhanced Raman scattering spectroscopy. The star (*) in front of “Signaling procedure” means a single-molecule fluorescence readout (ability to sense the fluorescence from a single DNA origami structure).

Signaling Procedure	Analyte	Enhancement Factor	Response Time	Publication Year	Reference
Hotspot (two AuNPs)	TAMRA	-	-	2013	[36]
Hotspot (two AuNPs)	SYBR gold (25 dyes)	1.4 × 10^5^	-	2014	[117]
*Hotspot (two AuNPs)	Rhodamine 6G, ssDNA	10^7^ (dye) 10^5^ (ssDNA)	-	2014	[116]
Hotspot (four AuNPs)	aminobenzenethiol (4-ABT)	10^2^/nanoparticle	-	2014	[124]
*Hotspot (two AuNPs)	Single Cy3.5	10^2^ (for gap 1.4 nm vs. 2.5 nm)	-	2016	[53]
*Hotspot (two AuNPs)	TAMRA and Cy3	10^10^	-	2016	[121]
Hotspot (two AuNPs + graphene)	TAMRA	-	-	2016	[120]
*Hotspot (two gold nanostars)	Single Texas red	2.0 × 10^10^ (particles gap of 7 nm) 8.0 × 10^9^ (particles gap of 13 nm)	-	2017	[118]
Hotspot (gold nanolenses)	TAMRA	1.4 × 10^6^	-	2017	[123]
*Hotspot (two gold nanoprisms)	Cy5 and Cy3	10^9^ to 10^10^	-	2018	[119]
Hotspot (silver nanolenses)	Streptavidin	10^1^ (in blue region, vs. gold nanolenses) 4.0 × 10^0^ (at 532 nm, vs. gold nanolenses)	-	2018	[125]
Single silicon nanowire	Methylene blue	1.1 × 10^5^	-	2019	[122]

**Table 5 materials-13-02185-t005:** Summary table of methods based on DNA origami structures using circular dichroism (CD) signal change as readout. Abbreviations: Left-handed molecules (LH), Right-handed molecules (RH), Adenosine triphosphate (ATP), proof of principle (POP), Open and close structure configuration change (O/C).

Signaling Procedure	Analyte	Sensitivity	Response Time	Publication Year	Reference
Measurement of CD signal	POP	-	-	2012	[133]
Measurement of CD signal	POP	-	-	2012	[129]
O/C structures	Fuel	70 nM (first cycle)	-	2014	[139]
RH, LH structures proportion	pH	Range depends on the percentage of LH or RH molecules	Few minutes	2017	[20]
O/C structure	Viral RNA (Hepatitis C virus)	100 pM (Buffer) 1 nM (Human serum)	30 min (incubation) 0.01 s (each wavelength)	2018	[19]
O/C structure	Adenosine	20 µM (Lock style 1) 65 µM (Lock style 2)	1 min (Lock style 1)	2018	[137]
O/C structure	ATP and Cocaine	mM to µM range (ATP or Cocaine)	-	2018	[138]
O/C structure	Human α-thrombin	100 pM	-	2019	[52]

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
