# Peer review of "DNA Origami as Emerging Technology for the Engineering of Fluorescent and Plasmonic-Based Biosensors"

_materials, 2020, doi:10.3390/ma13092185_

Round 1
Reviewer 1 Report
The review by Loretane et al. summarizes the state of the art of DNA origami based biosensing strategies with focus fluorescence based techniques and plasmonically enhanced optical spectroscopy. The field is developing fast, hence a concise review is very helpful. The present manuscript is well-composed and I generally recommend its publication in Materials, but the authors need to take into account the following issues:
- There are biosensing approaches, which rely on DNA origami and are very promising, but not mentioned by the authors. They should consider to discuss also the following papers: J. Am. Chem. Soc. 2017, 139, 46, 16510-16513, ACS Appl. Mater. Interfaces 2019, 11, 2, 2295-2301.
- In general the text needs to be carefully spell-checked (e.g. page 3 and further below: the authors repeatedly use the term "bimolecular sensing" - I believe they mean biomolecular), in particular, many articles are missing.
- Figure 1: It would be good to have an image of the assembled DNA origami that illustrates how it is made up from the colored staple strands and the black scaffold, the assembly principle is not made clear from the simple grey tubes on the right.
- The abbreviation M.L. is used for two authors, the author contributions ("contributed equally", etc.) therefore are ambiguous.
- The bibliography has two entries with #1, because of this, all reference numbers in the text are shifted by one number.
- Line 48: "increase" should be used of instead of "increment".
- Line 104: What is meant by "resonant"?
- Line 107: "concentrations" should be used instead of "concretions".
- Table 1: Systemic Lupus Erythematosus is the disease to be detected, not the analyte.
- Table 1: the column title "LOD" does not match several of the entries, perhaps the authors should use a more general term.
- Line 207: it could be confusing for the reader that fluorescence enhancement is to be explained in the SERS section. The way it is formulated now, the impression could arise that both phenomena are the same thing.
- Table 2: It is misleading to state the fluorescence decay time in a column where otherwise the response time of the sensor is given.
- Figure 4 B doesn't explain the principle of this DNA walker experiment very well.
- The long description in lines 331 - 348 could be aided by a Figure.
- Table 3: "DNA Target" is too ambiguous: is it ssDNA or dsDNA? In other rows this is specified.
- Lines 389 - 393: These first lines are slightly misleading in an introduction to SERS. As I understand it, these advantages refer to detection schemes where the Rayleigh scattering of the metallic nanoparticles themselves is measured, not to SERS, where the amplified Raman scattering of an analyte molecule is detected.
- Figure 5 A: It is misleading to leave out the coating strands on the NPs.
- Figure 5 B: Please cite the source of these spectra.
- Line 418: Actually, none of the SERS experiments on origami really "tuned" the gap distance, the origami doesn't allow the nm-precision that is required to do this for SERS (except, perhaps the thermal tuning by Simoncelli et al.). For high EF, researchers usually just try to reach the smallest gap possible, i.e., particles touching at their coatings.
- Line 433: The "extra modification" (graphene sheet) in the reference was not within the origami, but on top of the plasmonic assembly.
- Line 438: I believe the authors meant "three to four"?
- Line 439: What is meant by "different arrangement"? "Single molecule" should be replaced by "SERS" in line 438, not all papers referenced in line 439 were single molecule studies.
- Line 439: Daems et al. did not measure SERS. Instead, consider to cite:
x Pilo-Pais, M., Watson, A., Demers, S., LaBean, T. H., & Finkelstein, G. (2014). Surface-enhanced Raman scattering plasmonic enhancement using DNA origami-based complex metallic nanostructures. Nano letters, 14(4), 2099-2104.
x Heck, C., Kanehira, Y., Kneipp, J., & Bald, I. (2018). Placement of Single Proteins within the SERS Hot Spots of Self‐Assembled Silver Nanolenses. Angewandte Chemie International Edition, 57(25), 7444-7447.
- Table 4: For one example, the authors give the integration time, for the others they don't. Why?
- Table 4, row 5: It is a bit misleading to give this relative enhancement factor here. If the overall enhancement factor is not available, rather leave it out. This is a single molecule study with a very strong enhancement after all. (To a lesser degree this also applies to row 4.)
- Table 4: Typos: SYBR gold, TAMRA
- Table 4: Remove Daems et al. here as well (no SERS).
- Line 450: Not sure whether it is the right terminology to call a photon right/left handed...
- Table 5, row 5: Not the whole Hepatitis virus but just respective viral RNA was detected in this study.
- Line 522: The ordinary DNA origami is not stable in biological fluids, hence the necessity for stabilization methods...
- Line 529: The price for a Nusinersen (Spinraza) injection given here is the price demanded by the pharmaceutical supplier, which also reflects factors such as development costs, number of potential patients (rare disease) and, above all, is a market-based prize. The actual manufacturing costs are likely to be a fraction of the 125,000 $.
Reviewer 2 Report
The work is innovative and surely necessary in these and future days of the acute need in accelerating the sensing and detecting at bio-Nano-level and interaction. Being a review, the work can be beneficially completed with some useful links for further developments, such as 1) Enzyme logistic kinetics: (DOI:10.3390/molecules16043128) with spectroscopic applications; 2) Spectral Inverse Method: DOI: 10.3390/ijms131215925, with Raman spectra application. However, the DNA origami method, despite very seductive, should be further amended with some in situ induced interaction, eventually by bio-Nano-compatible materials in the tissue situ, so that avoiding the false observations in the in vitro conditions, due to so many intermediary operations – specific for the Nano-lab approach. Overall, an inspiring contribution that can be accepted with the above recommended insertions.
Reviewer 3 Report
Loretan et al. have prepared an extensive review on DNA origami-based fluorescent and plasmonic nanosensors. The review is well written, and it gives a rather balanced overview on the topic. The figures are clear and useful, however some of the figures have a lot of ample space (e.g. Figure 3). By recomposing the figures, there would be much more space that could be easily used for adding more content to the review. That is why I think the following comment in the end of the article "We apologize for not being able to cite all published work relevant to this topic due to the selective focus of the review and limited available space" is not valid. I think this statement should be removed. Moreover, there are a few minor things that need to be addressed. If the authors could revise the manuscript based on my suggestions, I would and will recommend publication in Materials. Please see my detailed comments below:
- The reference numbering is somehow off at times, the authors should be careful that the numbering in the text follows the bibliography in a consistent way. In addition, some of the references miss the article or DOI numbers (if they are not yet included into issues).
- Importantly, the hotspots can also be realized using heterogeneous particles, see e.g. Roller et al. Nat. Phys. 2017, 13, 761. In addition, in this context I would also mention the possibility to create various nanostructures made of different materials using DNA-based lithography, see e.g. Shen et al. Sci. Adv. 2018, 4, eaap8978.
- The authors describe a DNA nanovault by Grossi et al. (for example that one also misses an article number), and I think the follow-up study by Ijäs et al. ACS Nano 2019, 13, 5959 would make a nice addition to that discussion. Ijäs et al. used a bit different origami design and employed pH-latches for reversible opening and closing of the "capsule" instead of strand displacement reaction. This technique significantly reduced the response time of opening.
- The authors may add a few references on DNA -based photonic, plasmonic and bio applications at interfaces. They have already cited Wang et al. Chem. Soc. Rev. 2019, 48, 4892 & Bui et al. Adv. Opt. Mater. 2019, 7, 1900562, but I would also recommend to check e.g. Gopinath et al. Nature 2016, 535, 401; Shen et al. Langmuir 2018, 34, 14911 & Stassi et al. Nat. Commun. 2019, 10, 1690.
Reviewer 4 Report
The authors present a review of DNA origami sensors with optical readouts. They do a nice job introducing the technologies and characterizing the sensors into fluorescence based sensors with 1) a direct readout, 2) FRET, and 3) fluorescence-quenching as well as SERS and CD based nanosensors. The figures and tables in each section clearly explain how each mechanism works and characterize studies in terms of response time and function. A forward looking perspective discusses important concerns such as large-scale production and stability of the nanosensors. The review is well-written and fairly comprehensive in a growing field. A few minor comments:
- Reference numbers are off by one. It looks like this is probably because there are two reference #1s
- Line 58: “complimentary” should be “complementary”
- Be careful with “bimolecular” vs “biomolecular” throughout the article to ensure you’re describing things in the way you intend
- In general, I think the authors should specify and emphasize which studies are using an ensemble fluorescent readout versus single-molecule fluorescence
